# On the Need to Tell Apart Fraternal Twins eEF1A1 and eEF1A2, and Their Respective Outfits

**DOI:** 10.3390/ijms22136973

**Published:** 2021-06-28

**Authors:** Alberto Mills, Federico Gago

**Affiliations:** Department of Biomedical Sciences & “Unidad Asociada IQM-CSIC”, School of Medicine and Health Sciences, University of Alcalá, E-28805 Alcalá de Henares, Spain; alberto.mills@edu.uah.es

**Keywords:** elongation factor 1A, G proteins, moonlighting proteins, genetic disorders, gene mutations, post-translation modifications

## Abstract

eEF1A1 and eEF1A2 are paralogous proteins whose presence in most normal eukaryotic cells is mutually exclusive and developmentally regulated. Often described in the scientific literature under the collective name eEF1A, which stands for eukaryotic elongation factor 1A, their best known activity (in a monomeric, GTP-bound conformation) is to bind aminoacyl-tRNAs and deliver them to the A-site of the 80S ribosome. However, both eEF1A1 and eEF1A2 are endowed with multitasking abilities (sometimes performed by homo- and heterodimers) and can be located in different subcellular compartments, from the plasma membrane to the nucleus. Given the high sequence identity of these two sister proteins and the large number of post-translational modifications they can undergo, we are often confronted with the dilemma of discerning which is the particular proteoform that is actually responsible for the ascribed biochemical or cellular effects. We argue in this review that acquiring this knowledge is essential to help clarify, in molecular and structural terms, the mechanistic involvement of these two ancestral and abundant G proteins in a variety of fundamental cellular processes other than translation elongation. Of particular importance for this special issue is the fact that several de novo heterozygous missense mutations in the human *EEF1A2* gene are associated with a subset of rare but severe neurological syndromes and cardiomyopathies.

## 1. Introduction

Protein synthesis is one of the most sophisticated biochemical processes in living cells and comprises the same steps in Eukarya, bacteria and archaea: initiation, elongation and termination-ribosome recycling notwithstanding. These three phases are strictly regulated by an assortment of protein components that make this process considerably more intricate and complex in higher eukaryotes than it is in prokaryotes [1,2]. During the translation elongation step, the G protein known as the eukaryotic elongation factor 1A (eEF1A), in complex with GTP, binds to and delivers the aminoacyl-tRNA (aa-tRNA) to the A-site of the 80S ribosome. Following hydrolysis of the ester bond linking β and γ phosphates in GTP, the guanine nucleotide exchange factor (GEF) eEF1B interacts with eEF1A to release it from the ribosome in its GDP-bound state, which adopts a different conformation and is thought to be involved in other functions, despite the often-assigned appellative of being “inactive” [3,4]. tRNA charging onto eEF1A and delivery to the ribosome is a highly efficient channeled process that requires additional factors [5], which together make up the so-called eEF1 complex, whose composition changes as the complexity of the eukaryotic organism increases [6]. This high-molecular-weight complex has been characterized in *Artemia salina* [7] and human cells as a pentamer made up of two eEF1A subunits, two GEFs (eEF1Bα and eEF1Bδ) and the structural scaffold protein eEF1Bγ, which can bind to cytoskeletal components [8].

Remarkably, in B cells, 30–50% of the endogenous eEF1A exists as a dimer [9], and an aa-tRNA-free form of GDP-bound eEF1A retains the activation-induced deaminase (AID) enzyme in the cytosol [10]. Upon translocation to the nucleus, AID acts on the immunoglobin loci to cause U:G mismatches in the genomic DNA through deamination of deoxycytidines into uracil and thus trigger antibody gene diversification [11].

In a neuronal context, the mRNA encoding eEF1A is constitutively present in the dendrites of many neuron types [12] where its local translation is regulated by the Fragile X mental retardation protein (FMRP) [13] and modulated by synaptic activities [2,3,4,5,6,7,8]. This dendritic localization is reportedly highest when overall synapse density is low at postnatal day 7 [14] and less abundant at days 10 and 15, in sharp contrast to that of other dendritic mRNAs, which increase in abundance as neurons mature. This P7-P15 period is a time of extensive dendritic growth and rapid synaptogenesis via dendritic filopodia [15].

eEF1A mRNA, like those of many vertebrate ribosomal protein subunits, possesses a TOP sequence, i.e., an oligopyrimidine-rich tract, at the 5′ end, which ensures rapid translation. This is important in some brain areas, such as the hippocampus, because evidence has accumulated over the years that long-lasting forms of activity-dependent synaptic plasticity [16], such as long-term potentiation, which provides a cellular model for learning and memory [17], and long-term depression [18] of neurotransmitter release, require the local synthesis of proteins [19] within dendrites in the absence of new transcription [20]. Among the signaling pathways that regulate both translation initiation and elongation, activation of group I metabotropic glutamate receptors mGluR1 and mGluR5 stand out [21,22]. These receptors are positively coupled to phospholipase Cβ via G_q_ proteins and are typically activated in lab experiments by the agonist 3,5-dihydroxyphenylglycine, which can trigger dramatic enrichments in eEF1A upon local applications to dendrites due not only to new protein synthesis but also to a redistribution of existing eEF1A possibly related to a reorganization of the actin cytoskeleton [12]. Of note, dysregulation of eEF1A has been correlated with synaptic plasticity impairments in Alzheimer’s disease [23] and the mRNA encoding the microtubule-associated protein tau has been found associated with eEF1A in the axons of cultured rat cortical neurons at long distances from the cell body [24].

To further illustrate the far-reaching roles of eEF1A, the GEF eEF1Bα is efficiently displaced from preformed nucleotide-free complexes upon binding of ZPR1 to GDP-bound eEF1A. ZPR1 is a 2xC4 zinc finger protein that binds to subdomains X and XI of the cytoplasmic tyrosine kinase domain of the epidermal growth factor receptor (EGFR). In response to growth stimuli, ZPR1 assembles into complexes with eEF1A and the survival motor neurons (SMNs) protein [25] and accumulates in subnuclear structures (Cajal bodies and gems) [26]. The ZPR1 domain tandem has been shown by X-ray crystallography to consist of an elongation initiation factor 2-like zinc finger and a double-stranded β helix with a helical hairpin insertion. Treatment of mammalian cells with EGF leads to tyrosine phosphorylation in the cytoplasmic domain of the EGFR [27], release of ZPR1 from EGFR, direct binding of ZPR1 to eEF1A and ZPR1 accumulation in the nucleus [28]. The SMN1 protein, which associates with cytoskeletal elements in spinal dendrites and axons during postnatal development, has been implicated in axonal transport in the spinal cord [29].

## 2. eEF1A1 Versus eEF1A2

More than one version of eEF1A exist in many organisms. In mammals, two sister proteins named eEF1A1 and eEF1A2 are encoded in separate genes, *EEF1A1* and *EEF1A2*, which are located in different chromosomes, e.g., 6 and 20 in humans and 9 and 2 in mice, respectively [30,31]. *EEF1A1* is almost ubiquitously expressed whereas *EEF1A2* shows a very restricted pattern of expression, in common with an isoform of eEF1Bγ (formerly eEF-1β) [32]. Furthermore, eEF1A1 is the embryonic form of the elongation factor for protein synthesis whereas its sister, eEF1A2, is the adult form in neurons, myotubes and cardiomyocytes [33]. Minor sites of normal *EEF1A2* expression have also been found in several specialized human body compartments [34], including breast acini [35], glucagon-producing islet cells in the pancreas, Purkinje cells of the cerebellum [36], lung alveoli [37], duodenal Brunner’s glands and goblet cells in colon crypts [36].

eEF1A2 is known to lead to cytoskeletal remodeling and malignant transformation of a variety of mammalian cells [35,38,39] that may have in common the formation of three-dimensional grape-like clusters termed acini and a propensity to invade adjacent tissues and metastasize. Intriguingly, phosphatidylinositol (PI) 4-kinase IIIβ (PI4KB), an enzyme that localizes to the basal surface of acini created by normal human breast epithelial cells [40], disrupts mammary acinar formation in vitro whereas its ectopic expression induces multiacinar development, a phenotype that can be recapitulated by expression of *EEF1A2* [41]. eEF1A2 also promotes the formation of cytosolic projections (filopodia and lamellipodia) consisting of actin bundles [42] that are important for 3D cell migration in both healthy and diseased cell conditions [43]. In this respect, it is also of interest that the interaction between eEF1A and actin is critical for anchoring β-actin mRNA to the cytoskeleton [44].

Unfortunately, bibliographic sources have often referred, and still refer, to “eEF1A” without specifying whether the protein is actually eEF1A1, eEF1A2 or a mixture of the two. This scenario is currently changing, though, as interest in their simultaneous quantification [45] has grown rapidly in recent years because eEF1A2 has been shown to sustain the progression and aggressiveness of several cancers [46]. Nonetheless, obtaining biochemical and mechanistic data on eEF1A1 and eEF1A2 has been challenging because qualitative and quantitative discrimination of these fraternal twins was hampered for a long time due to (i) the ‘background noise’ imposed by their high abundance within cells (>1% total protein content) [47], (ii) the late advent of eEF1A1- and eEF1A2-specific antibodies (Table 1) [33,48] and (iii) a profusion of post-translational modifications (PTMs) [49] that provide them with matching or distinct outfits and further diversify their respective interactomes [50].

The fact that eEF1A1 and eEF1A2 are 92% identical and 98% similar (Figure 1) has led many scientists to believe that they are isoforms rather than paralogs but their presence in most normal cells is mutually exclusive and development-specific for reasons that are not well understood [51,52]. Interestingly, eEF1A2 proteins from different mammalian species are more related to each other than eEF1A1 and eEF1A2 from the same species are. Although the activities as protein elongation factors of eEF1A1 and eEF1A2 have been shown to be indistinguishable in an in vitro translation system, GDP dissociates 7-fold more slowly from eEF1A2 than it does from eEF1A1 [53]. This change in nucleotide binding affinity has been ascribed to the nature of the amino acid at position 197, which is an asparagine in eEF1A1 but a histidine in eEF1A2 [53] (Figure 1).

### 2.1. eEF1A1 and eEF1A2 Are G Proteins with Three Distinct Domains and Very Different Conformations in Their Monomeric and Dimeric Forms

eEF1A1 and eEF1A2 fold into three well-characterized structural domains, namely, domain I (residues 4–234), where GDP/GTP binding takes place, and domains II (241–328) and III (337–462/463), both of which are β-barrels. In the monomeric ‘GTP conformation’ of eEF1A1 [57], Asp110 is involved in a strong ionic interaction with Arg240, but this latter residue is fully exposed to the solvent when two GDP-bound eEF1A2 monomers are mutually associated, as found in the eEF1A2 dimer from rabbit skeletal muscle solved by X-ray crystallography [58,59]. In fact, in this dimeric form, each Arg240 is located in the first half of a proline-rich motif of sequence ^235^ILPPTRPTDKPLRLPL^250^ that is not fully extended but has been proposed to play a critical role in sequence recognition by the SRC homology 3 (SH3) domain of effectors such as phospholipase Cγ1 (PLCγ1) and adaptors such as Nck1 and Nck2 [59].

Careful examination of the monomer–monomer interface [59] revealed the presence and high relevance for dimer stabilization of Gly70, Ile71, Asp252 and Arg423 and also Tyr86 and Lys100, which are spatially close to Asp91 and Phe98. This profusion of amino acids that are not only strictly conserved throughout evolution [34], but also substituted due to missense mutations in children affected with severe neurodevelopmental disorders (Table 2), appears to support a role for the eEF1A2 dimer (Figure 2) in cellular functions that are necessarily distinct from that of delivering tRNA to the ribosome. Furthermore, the higher-order association of two eEF1A2 dimers in the crystal lattice gives rise to a tetramer in which the negatively charged carboxylates of Glu122 and Glu124 (also found in Table 2) interact (at the dimer–dimer interface) with the positively charged side chains of Arg166 and Lys154. In addition, Glu122 and Glu124 flank Phe123, a hydrophobic residue located at the tip of a loop that has been posited [59] as key for the interaction of the membrane-bound eEF1A2 dimer with PI4KB, a crucial enzyme (EC 2.7.1.67) in the biosynthesis of phosphoinositides that play essential roles in signal transduction, cytokinesis, regulation of lipid and protein transport from the Golgi complex to the PM and determination and maintenance of the identity of several cell organelles [40,60].

The variable positioning of domains II and III with respect to domain I is possible due to the existence of intrinsically disordered regions (IDR) between them. IDRs within proteins containing folded domains usually provide a new layer of regulation to their activities because the conformational variability is sensitive to small environmental changes within the cell and thereby allows the integration and modulation of cellular responses to complex inputs, as shown for Src family kinases [61,62,63]. In the case of eEF1A1 and eEF1A2, this conformational richness is emerging as a crucial balancing factor to differentiate between relatively independent functions of these sister proteins, for example, translation elongation and cytoskeletal organization. Consequently, it is reasonable to assume that eEF1A2 pathological variants are unable to fulfill all of these functions due to conformational limitations.

### 2.2. eEF1A1 and eEF1A2 ‘Moonlighting’

eEF1A1 and eEF1A2, in common with their prokaryotic EF1A (aka EF-Tu) counterparts [64], participate in a multitude of other less explored cellular functions that are collectively termed “non-canonical” [65]. It can be argued that this multitasking (aka “moonlighting”) ability largely relies on specific PTMs, which are likely to have an impact on conformational preferences [66,67] and subcellular localization [65] of the resulting proteoforms [68]. A case in point is the covalent attachment, via an amide bond, of phosphatidylethanolamine (PE) to glutamate residues Glu301 and Glu374 [69,70,71,72,73] by means of a poorly characterized but evolutionarily conserved [74] biochemical reaction akin to the conjugation to PE that takes place on autophagy proteins LC3 (in mammals) and Atg8 (in yeast) to promote their stable association with the autophagosome membrane [75,76]. Hence we can infer that a likewise modified eEF1A1/2 will follow suit, given the key role played by PE in membrane architecture [77]. This type of membrane attachment for eEF1A1/2 would conform with available experimental evidence supporting (i) their localization close to nuclear and plasma membranes [31,78,79], (ii) their anchoring at the ER membrane [80], (iii) the existence of eEF1A1–eEF1A2 interactions at the PM [30] and (iv) their established role in facilitating contacts with other membrane and cytoskeletal components [81].

## 3. Involvement of eEF1A2 in Genetic Disorders

eEF1A2 really came into focus in 1998 when it was reported that the *‘wasted’ (wst)* phenotype described in 1982 for some Jackson Laboratory’s inbred mice [82] was due to a de novo 15.8-kb deletion that resulted in the removal of the promoter and the first (noncoding) exon of the *EEF1A2* gene, thereby obliterating all presence of the eEF1A2 protein [51]. This autosomal recessive phenotype is characterized by progressive weight loss and muscle wasting (hence the name of the mutation), neurological defects and immunological abnormalities. Homozygous *wst/wst* mice are normal after birth but they develop progressive paralysis by day 21 and die one week later. Intriguingly, during this postweaning period, lymphoid cells, but not fibroblasts, display a defective response to γ ray-induced chromosomal damage [83]. This onset of severe symptoms is coincident with a genetic switch in wild-type (+/+) and heterozygous (+/*wst*) mice that silences *EEF1A1* expression and promotes *EEF1A2* expression in motor neurons of the spinal cord, brain neurons, but not astrocytes and microglia [36], cardiomyocytes and skeletal myotubes. In (*wst*/*wst*) mice, however, the equally programmed disappearance of the eEF1A1 protein [33,84] cannot be accompanied by the concomitant replacement with eEF1A2 and, consequently, the animals die before 30 days of age. Remarkably, restoration of eEF1A2 in muscle but not in neurons fails to rescue the wasted phenotype [36]. Strikingly, during apoptosis following caspase 3 activation induced by serum deprivation, the eEF1A2 protein disappears from dying myotubes and is replaced by eEF1A1 [85]. In this respect, cell death is delayed by enforced continuous expression of *EEF1A2* or transfection with antisense *EEF1A1* whereas it is accelerated by introduction of *EEF1A1*. It is thus clear that the two sister genes, *EEF1A1* and *EEF1A2*, regulate myotube and neuron survival by exerting prodeath and prosurvival activities, respectively [85].

In recent years, several de novo heterozygous missense mutations in the human *EEF1A2* gene (Table 2) have been associated with epilepsy and intellectual disabilities in children [34,86,87,88,89] that are often accompanied by specific facial dysmorphologies. Furthermore, some cases of these neurodevelopmental phenotypes were also presented with dilated cardiomyopathy and were found to be associated with both homozygous and heterozygous pathogenic EEF1A2 variants (p.P333L [90] and p.V16L [91], respectively). Since the most frequently occurring mutation in the *EEF1A2* gene from these patients is p.(G70S), the CRISPR/Cas9 genome editing technology was elegantly employed to create a breeding line of mice encoding this amino acid replacement in their *EEF1A2* gene. Although the high frequency of biallelic mutations in *EEF1A2* meant that none of the mice carrying the missense mutation survived past 4 weeks, as expected, and high levels of non-homologous end-joining DNA repair prevented the recapitulation of the clinically relevant human G70S/+ genotype, many of the F0 founder mice developed motor neuron degeneration whereas others displayed phenotypes consistent with severe neurodevelopmental disorders. Since a (G70S/G70S) mouse turned out to be more severely affected than (−/−) mice and the presence of G70S eEF1A2 was insufficient to protect (G70S/−) mice from neurodegeneration, it was concluded that this protein variant is essentially non-functional and, consequently, there must be both a loss of function and a dominant negative/gain of function at play in the children with missense mutations in *EEF1A2* [92]. More recently, knocking down *EEF1A2* expression in zebrafish, using translation blocking and splice-site interfering morpholinos, showed that EEF1A2 deficiency in this species leads to skeletal muscle weakness, cardiac failure and small heads, a morphant phenotype that could be rescued by the use of human EEF1A2 wild-type mRNA, but not mutant RNA [90].

## 4. Post-Translational Modifications in eEF1A1/2

In resting cells at low pH, the majority of eEF1A (70–90%, depending on cell type) is bound to cytoskeletal F-actin and is therefore inactive in protein synthesis [44]. A physiological pH increase promotes the GTP-dependent binding of eEF1A to aa-tRNAs, thus changing its spatial distribution and upregulating protein translation [94]. This dynamic functional switch is likely to be driven by a conformational transition brought about by what can be considered [95] a PTM by protons, i.e., an alteration in the charge of crucial amino acid side chains (e.g., Asp35, Glu45, Glu68, Asp91, His95, Asp252, Asp362 and His367 [59]). Likewise, other PTMs can be seen as a source of additional functional richness and diversity in eEF1A1 and eEF1A2 [49] that enable these ubiquitous proteins to interact with other molecular partners in other cellular processes different from translation. These PTMs (leaving apart lysine ubiquitylation and sumoylation) can be grouped as follows:

### 4.1. PE Attachment to Glu301 and Glu374

This unique PTM of eEF1A1/2 at Glu301 in domain II and Glu374 in domain III, despite being of general knowledge [70,71,72,96,97] and deemed to be absolutely required for attachment to particular membrane domains [52,53], via insertion of the palmitate chain into a lipid monolayer [59], has not been observed or reported in any crystallographic structure published to date. Nonetheless, recent crystallographic and mass spectrometry (MS) results have shown that the side chain carboxylates of both Glu301 and Glu374 of eEF1A2 from rabbit skeletal muscle are covalently bonded to ethanolamine phosphoglycerol (EPG), a deacylation product of the PE molecule [59]. Of note, when the palmitate chains are attached to the EPG moieties bonded to Glu301 of both monomers, the fatty acids are oriented in a parallel fashion that defines a feasible mode of anchoring eEF1A2 to cell membranes (Figure 3A). Strikingly, the similarly PE-modified Glu374 residues can stick out at right angles from this dimer (or in parallel, depending on the conformational preferences of the phosphoglycerol moiety), but when a tetramer is considered, a different dimerization interface emerges and a pairwise parallel arrangement of phospholipids is also found on opposite sides of the tetramer (Figure 3B). These structural cues pave the way for future experimental and modeling studies that will shed more light on the intricacies of eEF1A1/2–membrane interactions.

### 4.2. Serine and Threonine Phosphorylation

In the two dimeric structures of eEF1A2 independently solved by X-ray crystallography to date [58,59], the hydroxy group of Ser21 is hydrogen bonded to the terminal β-phosphate of the bound GDP molecule. This serine is a well-known Raf-mediated phosphorylation site in both eEF1A1 and eEF1A2 that plays a crucial role in reducing the half-life of both proteins because it affects their ubiquitination and subsequent proteasomal degradation. Both Ser21 and Thr88 of eEF1A1 have been shown to be phosphorylated by B-Raf but not C-Raf in vitro whereas Ser21 of eEF1A2 was reported to be phosphorylated by both B- and C-Raf [98]. Intriguingly, the finding that Ser21 phosphorylation is strongly enhanced when both eEF1A1 and eEF1A2 are preincubated together prior to the assay with C-Raf hints at increased accessibility of Ser21 to the phosphate following heterodimerization, a possibility that is particularly evident at the PM [30]. Nonetheless, in the reported eEF1A2 dimer structure, neither Ser21 nor Thr88 (whose hydroxy group is involved in a hydrogen bond to the carboxylate of Asp35) is phosphorylated.

Ser53 undergoes a well-established Ca^2+^-activated, PKCβ1-dependent [28] phosphorylation. Remarkably, this residue is also targeted by *Legionella pneumophila* glucosyltransferases Lgt1 and Lgt2, when eEF1A is loaded with GTP and aa-tRNA (Figure 3C) [27], to inhibit protein synthesis. In the X-ray crystal structure of eEF1A2, the hydroxy groups of the bound GDP ribose are solvent-exposed and face the 50–56 loop from another monomer in the crystal lattice so that four different nucleotide environments are discernible in the crystal lattice, of which only three would be compatible with Ser53 phosphorylation or glucosylation. The high mobility of this region allows for the alternate and mutually exclusive positioning of the side chains of Ser53 and Phe54 [59].

TGF-β-mediated phosphorylation of Ser300 in eEF1A1 prevents aa-tRNA binding [99] and lead to downregulation of mRNA translation and cell proliferation [99,100]. These experimental findings are in consonance with the presence of a phosphate group attached to Ser300 in the recently reported X-ray crystal structure of eEF1A2 from rabbit skeletal muscle in contrast to the previously published phosphorylation of Ser163 and Thr239 on the same protein [58]. None of these structures, however, showed any strong evidence of glycosylation, despite the fact that both Ser300, preceding in the primary sequence one of the glutamate residues that is covalently bonded to PE, and Thr239 of eEF1A2 are predicted by the OGTsite method [101] as residues with a high likelihood (scores of 0.7/1 and 0.8/1, respectively) of having their side-chain hydroxyl groups β-linked to *N*-acetylglucosamine (GlcNAc) units. We believe this is of interest because, in many proteins, there is direct competition between *O*-GlcNAcylation and *O*-phosphorylation for the same Ser/Thr residues. Moreover, there can be adjacent or multiple occupancies for phosphorylation and *O*-GlcNAcylation on the same protein [102]. The interplay between these two PTMs creates further molecular diversity and is used to fine-tune protein expression, degradation and trafficking, and to regulate signaling events [103,104]. In this respect, the eEF1A1 peptide ^220^DGNASGT^226^, which is present in a solvent-exposed antiparallel β-sheet of domain I, was found in HEK293 cells to be *O*-GlcNAcylated (with a positive predictive value of 0.9 over 1.0) [105] but no crystallographic or MS evidence of Ser modification in the equivalent ^220^EGNASGV^226^ sequence of rabbit skeletal muscle eEF1A2 has been found [58,59].

Ser205 and Ser385 were shown, in HEK293T and HEK293E cells, to be phosphorylated by stress-activated c-Jun N-terminal kinase, following recruitment by the 40S ribosome-associated RACK1 (receptor for activated protein C kinase 1) to promote degradation of nascent polypeptides by the proteasome [106]. In addition, recent work using phosphoablated and phosphomimetic eEF1A2 variants (i.e., Ser→Ala and Ser→Glu at positions 342, 358, 393 and 445, respectively) has elegantly demonstrated that the configuration of phosphorylation sites in eEF1A2 modulates actin dynamics and structural plasticity during dendritic spine remodeling in neurons [107]. Thus, in stable spines, a subpopulation of non-phosphorylated eEF1A2 would be involved in translation as monomers whereas dimers would participate in bundling F-actin. Interestingly, the X-ray crystal structure of eEF1A2 from skeletal muscle does not reveal any extra electron density suggestive of phosphorylation at Ser205 and Ser385 [58,59], but it shows a feasible mode of membrane-bound dimerization that places Ser385 from both monomers in close apposition and each facing the hydroxyl group of Ser354 from the other monomer (Figure 4). This arrangement invites speculation that Ser358 phosphorylation could strengthen this type of dual interaction and contribute to dimer stabilization.

### 4.3. Tyrosine Phosphorylation

Src Homology 2 (SH2) domains are sequence-specific phosphotyrosine-binding modules that are involved in protein–protein recognition and take part in multiple signaling cascades. Although binding of eEF1A2 to the SH2 domains of Grb2, RasGAP, Shc and the C-terminal region of phosphatase Shp2 has been demonstrated [108] and several Tyr residues (29, 85, 86, 141, 162, 167, 254 and 418) in eEF1A2 have been reported to be phosphorylated, the particular tyrosine(s) involved in each case remain(s) unknown. By carefully inspecting the surroundings of the side chains of all Tyr residues in dimeric eEF1A2, we rule out any phosphorylation on Tyr29, Tyr85, Tyr86, Tyr162, Tyr167 (in the ^165^KRYDEIV^171^ stretch), Tyr254 and Tyr418 because of the proximity of the phenolic OH to carbonyl or carboxylate oxygens. In fact, (i) phosphorylation on Tyr29 has been associated to stabilization of α-helices A-A’ and impairment of translation [109]; (ii) eEF1A1 is known to undergo structural rearrangements upon phosphorylation of Tyr85 and Tyr86 [110]; (iii) Tyr141 phosphorylation has been related to the maintenance of an extended conformation of eEF1A1 [110]; and (iv) a phosphorylated Tyr418 is known to modulate the interaction with mRNA and other binding partners [111]. The only solvent-exposed tyrosine in the structurally characterized eEF1A2 dimer is found at position 141, in the ^138^LLAYTLG^144^ amino acid stretch that is predicted by the eukaryotic linear motif resource [112] as a STAT5 SH2 binding motif. Nonetheless, no evidence of phosphorylation was found (although a sulfate ion is close to the phenolic oxygen of Tyr141 in both chains) and both its location in an α-helix and its closeness to the amino groups of Lys146 and Lys439 seem to be incompatible with binding to an SH2 domain if the protein adopts this dimeric form. 

### 4.4. Lysine Acetylation

This charge-depriving PTM is widespread as a regulatory mechanism outside the nucleus [113,114,115,116] and proteome-wide analysis of lysine acetylation in *Saccharomyces cerevisiae* supports its broad regulatory scope [115]. eEF1A in *Toxoplasma gondii* is known to be acetylated at seven independent lysine residues [114] and the human homologues are also heavily acetylated [116]. In whole-cell lysates of human acute myeloid leukemia MV4-11 cells, the lysines reported to be modified by acetylation were 172, 318, 392, 395 and 439 in eEF1A1 and 41, 44, 146 and 255 in eEF1A2 [116]. These findings strongly suggest an important, conserved role for lysine acetylation in the activities of eEF1A1 (https://www.phosphosite.org/proteinAction.action?id=3315, (accessed on 18 May 2021)) and eEF1A2 (https://www.phosphosite.org/proteinAction.action?id=3316, (accessed on 18 May 2021)) [117] that deserve further studies. Intriguingly, increased lysine acetylation of eEF1A has been observed in human hepatoma PLC5 cells upon treatment with the antitumor drug etoposide [118].

### 4.5. Lysine Mono-, Di- and Trimethylation

eEF1A is the eukaryotic protein that can be modified by the highest number of independent methyltransferases [119,120]. This family of enzymes [121] catalyzes the transfer, using S-adenosyl-L-methionine (AdoMet) as the cofactor [122], of up to three methyl groups to the lysine ε-amino group and also to the N-terminal site of eEF1A following methionine removal [123]. Dimethyllysine at residues 55, 165 and 290 and trimethyllysine at residues 36, 79 and 318 have been previously identified by MS in eEF1A1 purified from rabbit reticulocytes. METTL13 (methyltransferase-like 13) dimethylation of eEF1A on Lys55 (eEF1AK55me2) is utilized by Ras-driven cancers to increase translational output and promote tumorigenesis in vivo. METTL13-catalyzed eEF1A methylation increases eEF1A’s intrinsic GTPase activity in vitro and protein production in cells. METTL13 and eEF1AK55me2 levels are upregulated in cancer and negatively correlate with pancreatic and lung cancer patient survival. METTL13 deletion and eEF1AK55me2 loss dramatically reduce Ras-driven neoplastic growth in mouse models and in patient-derived xenografts from primary pancreatic and lung tumors [124], in line with findings in AML-193 and Kasumi-1 cells supporting the implication of eEF1A2K55me2 in acute myeloid leukemia [125]. METTL13 contains two distinct catalytic domains targeting the N terminus and Lys55 of eEF1A, respectively. Loss of METTL13 function alters translation dynamics and results in changed translation rates of specific codons [126], but other possible biochemical and functional consequences have not been explored in comparable detail. In fact, the side chain of Lys55 directly interacts with the indole ring of Trp196 in both GTP- [127] and GDP-bound forms [58,59] of eEF1A1/2, but only in the GDP-bound dimeric state does it interact with the phenol of Tyr56. Since we have pointed out that motions of Phe54, Tyr56 and Trp196 appear to be involved in nucleotide binding and exit [58], it is therefore conceivable that dimethylation of Lys55 serves the purpose of strengthening cation–π interactions [128] with the aromatic residues of Trp196, Tyr56 or even the neighboring Tyr29, to stabilize a particular protein conformation.

Human METTL21B specifically targets Lys165 in eEF1A when loaded with both aa-tRNA and GTP [129] whereas mammalian METTL10 (the human ortholog of yeast Efm4) specifically trimethylates eEF1A1 at Lys318 (equivalent to Lys316 in yeast) [130]. In this context, it is worth bearing in mind that chick neural crest cells undergo a protein methylation-dependent [131] epithelial-to-mesenchymal transition that allows them to migrate from the neural tube and extend lamellipodia and filopodia that promote targeting of other neurons, cardiomyocytes or glial cells in different body regions, such as peripheral nerves, heart, gut or skin [73,131,132]. Interestingly, eEF1A and S-adenosylhomocysteine hydrolase, which hydrolyzes the feedback inhibitor of *trans*-methylation reactions, are essential proteins in this process. In fact, replacing six Lys residues known to undergo methylation with alanines blocks neural crest migration altogether [131]. Together with results from studies in yeast, it becomes apparent that dynamic eEF1A1 methylation is dispensable for translation-related activities but necessary for engagement in crucial protein–protein interactions that are required for local actin translation, nucleation of actin polymerization, β-actin mRNA binding and targeting, and regulation of cell migration [131,133].

Regrettably, none of these lysine modifications could be unambiguously confirmed or negated in the recently solved X-ray crystal structure of eEF1A2 because of the high flexibility of these mostly exposed side chains (high B factors) and the limited data resolution (2.7 Å), which leads to voids in the corresponding electron density map regions. These shortcomings notwithstanding, the structural comparison between monomeric eEF1A1 in the GTP conformation (PDB entries 1SYW (obsolete PDB entry because it was a theoretical model, now available from ModelArchive (https://www.modelarchive.org/ (accessed on 13 May 2021). doi:10.5452/ma-cd28x)) [127] and 5LZS [57]) and each eEF1A2 monomer making up the dimer in PDB entry 6RA9 (as well as the superposition of individual domains from both proteins) reveals some notable differences originating from Lys36 and Lys79 that are compatible with lysine trimethylation. Thus, the strong hydrogen bond formed in eEF1A1(GTP) between the amino group of Lys36 and the carboxylate of Glu43 (held in place with the aid of a hydroxyl from Tyr29) is lost in eEF1A2(GDP) because Lys36, located at the N-terminus of an α-helix that is stabilized by the carboxylate of Asp35, is exposed to the solvent (and not involved in cation–π interactions with the spatially close rings of Tyr29 and Tyr56 in this protein conformation). Likewise, the hydrogen bonds from the ammonium group of the highly conserved Lys79 in eEF1A1(GTP) to the side-chain acceptor atoms of Thr88, Met294 and Glu297 are lost in the eEF1A2(GDP) dimer because the side chain of Lys79 is found in a completely different environment. In fact, its alternative location in this dimer raises the possibility of cation–π interactions [128] being established with the phenol ring of Tyr86 or with other binding partners.

### 4.6. Arginine Mono-ADP-Ribosylation (MARylation)

In proteins, the arginine side chain can be a substrate for enzymatic methylation, phosphorylation, citrullination and ADP-ribosylation reactions [134]. The extra electron density next to the N^ω^ atom of Arg266 in the GDP-bound eEF1A2 dimer purified from rabbit skeletal muscle [59] could not be ascribed to any of these PTMs but was found to be compatible with a carboxymethyl moiety, which could be present as a non-enzymatic advanced glycation end (AEG) product [135]. AGEs result from spontaneous hydrolysis of the hydroimidazolone adduct formed upon the Maillard reaction of the guanidinium group with glyoxal, an acyclic α-oxoaldehyde that is generated as a common autoxidation product in the metabolism of glyceraldehyde and monosaccharides like glucose and fructose [136,137]. This arginine modification appears to weaken the attraction to the carboxylate of Glu268 in domain III [58] and seems to be incompatible with the strong ionic interaction between Arg266 and the carboxylate of Asp110 in domain I that is observed in the monomeric GTP-bound conformation of eEF1A1 [127]. In this regard, the highly conserved Arg266 in mammalian eEF1A1/2 is present in a ^262^VPVGRVETG^270^ stretch that is similar to (in terms of flexibility), but distinct from (in terms of amino acid composition), the Gx(O/S)GER motif of collagen peptides (where O is hydroxyproline). This motif has been reported as the recognition site for collagen-binding integrins such as α_2_β_1_ [136,137]. Although hypothetical at present, this parallelism might be relevant because of (i) the prominent role played by integrins in tumor growth, metastasis and aggressiveness [138] and (ii) the fact that eEF1A acts as a membrane receptor for the cryptic antiadhesive site of fibronectin, whose proteolytic exposure has been linked to β_1_-integrin inactivation and subsequent apoptosis due to a loss of cell anchorage (“anoikis”) [139]. 

Mammalian cells have a large repertoire of Arg-specific ADP-ribosyltransferases (ARTs), which include the well-characterized Arg-specific cholera toxin-like (ARTC) and diphtheria toxin-like (ARTD) subclasses [140]. Recent in vitro profiling of the MARylomes of ARTD10 and ARTD11 in HEK 293T cell lysates identified eEF1A1 as a common target for both enzymes [141]. We ascertained the feasibility of a single ADP-ribose (ADPr) unit being transferred from the NAD^+^ cofactor onto Arg266 of each eEF1A2 monomer using molecular modeling. The side chains of both Arg266 residues appear to be equally accessible in the dimer (Figure 5A) and are found close to the N-terminus for which there is not enough electron density for unambiguous determination. This reversible PTM would entail (i) an increment in molecular mass of 2x540 Da, (ii) a size and length increase of the side chain and (iii) the addition of two formal negative charges from the phosphates to the single positive charge of the guanidinium group (at physiological pH values) [142]. Therefore, MARylation of eEF1A1/2 is likely to have a strong influence on both protein conformation and selection of the binding partners that recognize this particular region. In this regard, we point out that (i) the related toxin-catalyzed ADP-ribosylation of actin at Arg177 [143] sterically blocks actin polymerization [142] and (ii) the mosquitocidal toxin MTX produced by *Bacillus sphaericus* ADP-ribosylates *Escherichia coli* EF1A intensely, which results in bacterial protein synthesis inhibition because of the inability to form a stable ternary EF1A:GTP:aa-tRNA complex (although the targeted amino acid residue remains unknown) [144]. The introduction of the bulky ADPr moiety in eEF1A1/2 could sterically block the interaction with a binding partner, create a new docking site for ADPr-binding domains in other proteins or induce a conformational change [142].

Having established the likelihood and direct consequences of MARylation of Arg266, it did not escape our notice that each Arg266 side chain is spatially very close to Lys5, His7, Lys84 and Tyr86 of the same monomer and His295 and His296 from the other monomer. This cluster of residues could well make up, upon deprotonation of these histidines [145] at a pH above the 5.6 value used in crystallization, a poly(ADP-ribose)-binding zinc finger (PBZ)-like domain for MAR recognition, as found in APLF (aprataxin PNK-like factor) in complex with ribofuranosyladenosine (PDB entries 2KQD and 2KQE) and the E3 ubiquitin-protein ligase CHFR (checkpoint protein with FHA and RING domains, PDB ID: 2XOC). This attractive possibility could be very relevant for two main reasons. First, ADPr-Arg can be processed into secondary PTM, e.g., phosphoribosylarginine (the existence of which in the crystal structure of the eEF1A2 dimer cannot be ruled out due to disorder), or undergo the removal of the entire ADP-ribose moiety by ADPr-Arg-specific hydrolases [146,147], in which case the target protein function is restored [142]. Second, it is known that mammalian ARTC1 (encoded in the *ART1* gene, and first characterized in rabbit skeletal muscle) is GPI-anchored to the ER membrane and becomes activated during the ER stress response to target Grp78 (aka BiP and HSPA5). Interestingly, the acute ADP-ribosylation of Grp78 runs in parallel to the ensuing translational inhibition [148]. Thus, the ADP ribosylation of Grp78 on Arg470 and Arg492 has emerged as (i) a rapid posttranslational mechanism for reversible inactivation and (ii) an efficient strategy to minimize both aggregation and costly degradation of unfolded proteins [149], two essential processes in which eEF1A proteins are known to participate too [146,147].

In light of these considerations, it is remarkable that a recent genetic study reported that the first three patients diagnosed with a pathogenic p.R266W eEF1A2 variant showed marked clinical heterogeneity [88].

### 4.7. S-nitrosylation, S-glutathionylation and Disulfide Bond Formation

Of the six cysteine residues present in eEF1A1, only one (Cys234) is replaced by threonine in eEF1A2 (Figure 1). This particular cysteine, which is located spatially close to Cys111 in the eEF1A2 dimer, is a target site for S-nitrosylation in human cells [150]. Cys234 is also essential for dithiothreitol-sensitive oligomerization of eEF1A1 under oxidative stress conditions [56]. The fact that replacement of Thr234 in eEF1A2 by Cys234 promotes its oligomerization upon treatment with hydrogen peroxide supports the idea that eEF1A2 is intrinsically less prone to concatenation by this means. Reactive oxygen species also promote the formation of disulfide bonds between eEF1A1/2 and other redox-sensitive proteins. One relevant example is thioredoxin, which is part of the regulatory complex of the human 26S proteasome for which the elongation factor acts as a recruiter of misfolded nascent proteins for degradation, as shown in *D. discoideum* [151] and HeLa cells [152]. In addition, the patterns of intermolecular disulfide bonds involving eEF1A1 in yeast [153] and mammalian cells [154] upon exposure to various oxidative conditions can be modulated by varying the ratio of reduced to oxidized glutathione [154]. Remarkably, Cys411 in both eEF1A1 and eEF1A2 is known to be glutathionylated in response to glucose starvation [155] and this residue in the eEF1A2 dimer is found spatially close to Pro333, an amino acid that is replaced by Leu in some children suffering from neurodevelopmental disorders (Table 2).

### 4.8. α-N-Terminal Methylation

Following methionine removal, the N-terminal site and the adjacent lysine in yeast and human eEF1A have been shown to undergo a high-stoichiometry methylation that renders a trimethylated Gly2, and thereby a positively charged betaine, at the new N-terminus and a dimethylated Lys3 [123]. Unfortunately, electron density corresponding to the N-termini was lacking in the X-ray crystal structure of eEF1A2 for both monomers [58,59], but model building (Figure 5B) shows the proximity of this region to the loop embedding Ser300 and Glu301 and also to Arg266, three residues that are known to undergo other types of PTM. 

### 4.9. Methyl Esterification at the C-Terminus

The C-terminal region of eEF1A2 has been recently considered [59] as both an IDR and a bona fide WH2 motif with the ability to undergo a disorder-to-order transition upon binding to actin, in common with other actin-binding partners [156]. Interestingly, this conformational change involves a detachment from domain II and a change in orientation relative to domains II and III of the resulting α-helix. Moreover, since a methyl ester has been found on the α-carboxyl group of the C-terminal lysine residue of eEF1A from *Saccharomyces cerevisiae* [157], it seems reasonable to speculate that this is also present at the terminal lysines of eEF1A1 and eEF1A2. Nonetheless, the corresponding methylcarboxylated C-terminus of each protein is likely to make distinct subsite contacts with actin [59] because eEF1A1 lacks the penultimate glycine residue present in eEF1A2 (Figure 1). This distinguishing feature may, in turn, (i) account for the experimental observation that the resulting actin bundles are macroscopically different [158] and (ii) be important because colocalization of eEF1A1 with F-actin in lamella and filopodia has been detected in the cytoplasm of chicken neurons [131].

## 5. Final Remarks and Perspective

There is still much to be learnt about the mechanism(s) responsible and the underlying demands for switching off *EEF1A1* expression and switching on expression of the paralogous gene *EEF1A2* during myogenic differentiation and neuronal development [51,85]. So far, explanations based on protein synthesis [159] or putative structural variations between the monomeric forms [160], which differ in only a very small number of amino acids, have met with limited success.

Rabbit skeletal muscle has been used as an easily accessible natural source of eEF1A2 for X-ray crystallographic studies [58,59]. These have revealed just a limited subset of the large number of PTMs reported for this protein and/or its sister protein eEF1A1. Therefore, we are confident that bioinformatic analyses and molecular modeling can be combined with the experimentally obtained structural information to gain insight into some of the macromolecular ensembles in which eEF1A1/2 are involved, both in the cytoplasm and at specific membrane locations. In the context of neurodevelopment disorders caused by missense mutations in the *EEF1A2* gene, we believe that the reported 3D models, which account for the interaction with cellular membranes, F-actin and PI4KB [59], are particularly relevant in view of the essential role played by PI(4,5)P_2_ at neuronal synapses for coupling exo- and endocytosis [161] and as a source of the second messenger inositol trisphosphate (IP_3_) that is released from PI(4,5)P_2_ by the catalytic action of PLCβ and PLCγ.

We hope that this short review will epitomize attempts to bridge different scientific fields with common players and improve our understanding of the distinct roles played by human eEF1A1 and eEF1A2 in health and disease.

## Figures and Tables

**Figure 1 ijms-22-06973-f001:**
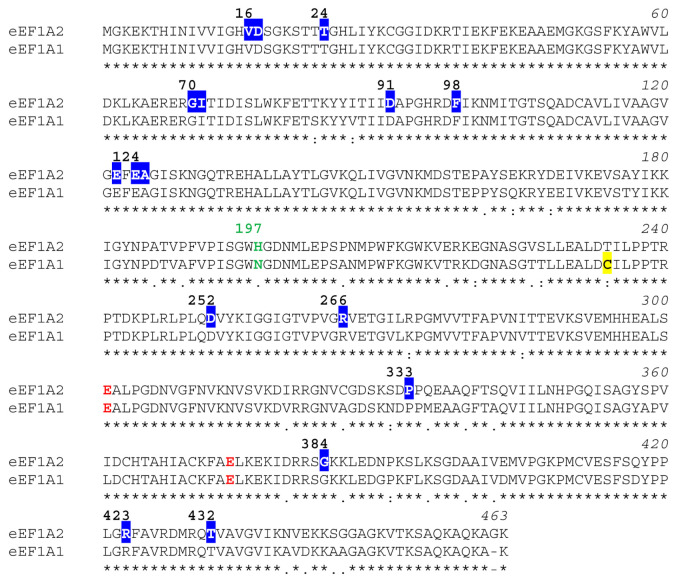
Sequence alignment of human eEF1A1 (EF11_HUMAN) and human eEF1A2 (EF12_HUMAN), which is 100% identical to rabbit eEF1A2 (mouse and rat eEF1A2 proteins only differ in one amino acid: Ser/Asn331Ala). The protein sequences were obtained from UniProtKB [54] and the alignment was produced with Clustal Omega [55]. Asterisks, colons and dots designate identical residues, conserved substitutions and semiconserved substitutions in the aligned sequences, respectively. The two Glu residues identical to those undergoing EFG modification in rabbit eEF1A2 (Glu301/Glu374) are colored in red. Cys234 in eEF1A1, which is essential for oligomerization) [56], is highlighted in yellow. Amino acids that are replaced upon missense mutations in the human *EEF1A2* gene and have been associated with severe neurological disorders are boxed and numbered. The amino acid at position 197 is colored in green.

**Figure 2 ijms-22-06973-f002:**
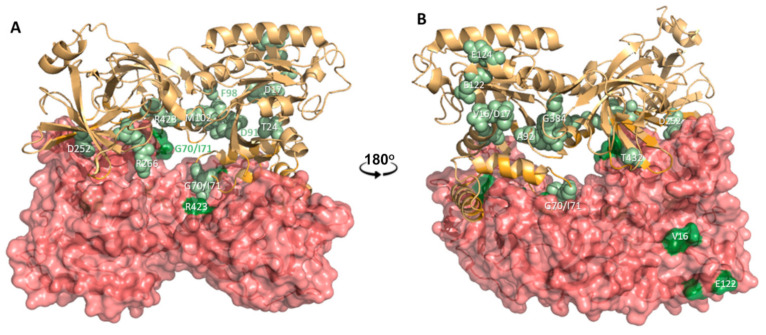
Cartoon representation of the crystallographic eEF1A2 dimer present in the asymmetric cell unit [59], with each monomer in a different color and one monomer enveloped in a van der Waals surface. Views in (**A**,**B**) are related by a 180° rotation about the Y-axis. The amino acids enumerated in Table 2 because of their known involvement in genetic disorders are colored in green and labeled in one-character notation. Note that the majority of these amino acids cluster at the dimerization interface and also between domains I and III of each monomer.

**Figure 3 ijms-22-06973-f003:**
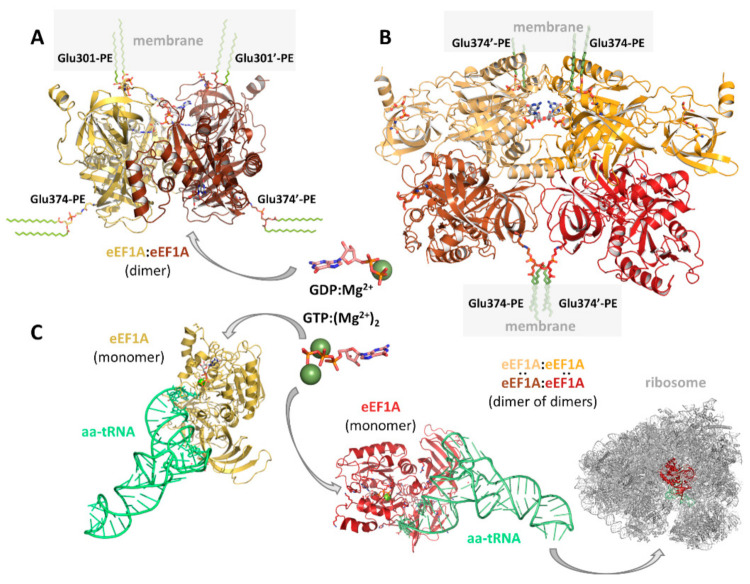
Scheme depicting theoretical 3D models of (**A**) a full-length post-translational modified eEF1A2 dimer with the PE chains attached to Glu301 of both monomers inserted into a simplified representation of a cellular membrane; (**B**) a dimer of dimers, as present in the X-ray crystal structure [59], showing the opposite orientation of the PE chains attached to Glu374 in the alternative pairs of dimers and their putative insertion at membrane contact sites; and (**C**) two eEF1A2:aa-tRNA complexes on their way to the ribosome (PDB entry 5LZS [57]). Note that the perpendicular orientation of PE molecules attached to Glu301 and Glu374 shown in (**A**) is rather arbitrary because of the rotational freedom of the phosphoglycerol moieties. The GDP and GTP nucleotides in the middle of the figure are displayed as sticks and the bound Mg^2+^ ions are shown as green spheres.

**Figure 4 ijms-22-06973-f004:**
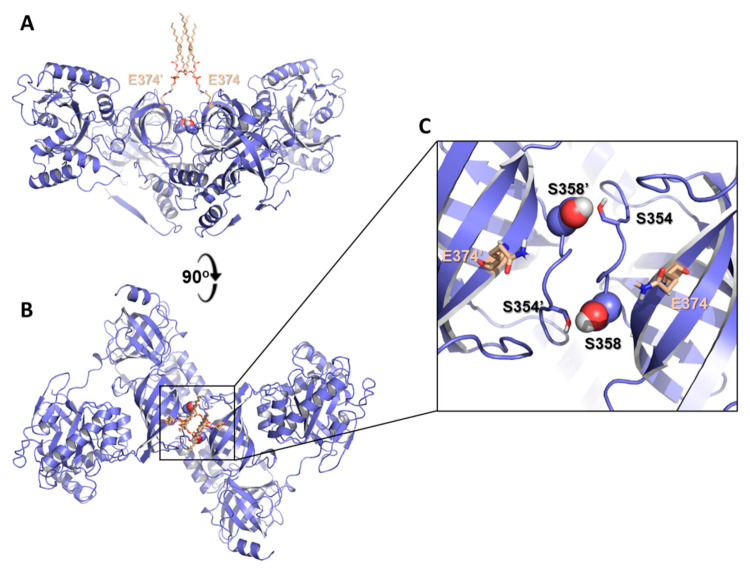
Side-by-side arrangement of two eEF1A2 monomers, each displaying a PE molecule covalently bonded to Glu374 (cream color), as found in the crystal lattice of the eEF1A2 structure solved by X-ray diffraction [58,59]. Views in (**A**) (equivalent to the upper part of Figure 3B) and (**B**) are related by a 90° rotation about the X-axis. The view enlarged in (**C**) shows the location of Ser354 (side chains as sticks) and Ser358 (side chains as spheres) in both monomers.

**Figure 5 ijms-22-06973-f005:**
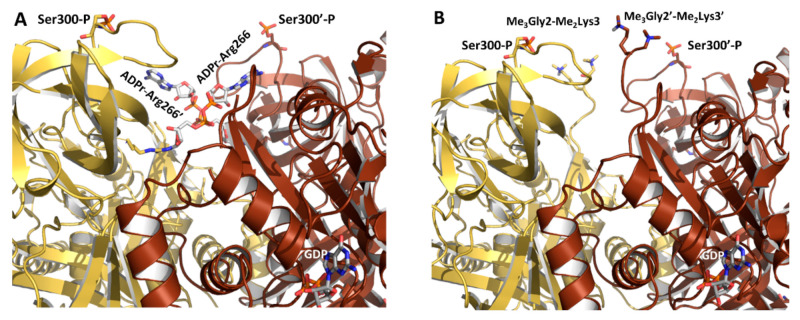
Theoretical models of (**A**) ADP-ribosylation on Arg266 and (**B**) N-terminal methylation on Gly2 and Lys3 of both monomers making up the eEF1A2 dimer, as present in the X-ray crystal structure [59]. Note the spatial proximity of both residues to Ser300 and Glu301, which also undergo other PTMs. The view is identical to that shown in Figure 3A. Ser300 is shown in a phosphorylated form (sticks) for reference purposes only.

**Table 1 ijms-22-06973-t001:** Peptide sequences used to generate specific antibodies against eEF1A1 and eEF1A2 ^a^.

eEF1A1	^326^AGDSKNDPPMEAAGFTAQ	^443^KKAAGAGKVTKSAQKAQKAK	**Year/Ref.**
eEF1A2	^326^CGDSKADPPQEAAQFTSQ	^443^KKSGGAGKVTKSAQKAQKAG	2001 [33]
eEF1A1	^439^KAVDKKAAGAGKVTC C^161^PYSQKRYEEIVKEVST ^217^TRKDGHASGTTLLEALDC		2007 [36]
eEF1A2	^439^KNVEKKSGGAGKVTC C^161^AYSEKRYDEIVKEVSA C^217^ERKEGNASGVSLLEALDT		

^a^ Amino acids differing between both paralogs are colored in red. The superscript denotes the eEF1A1/2 amino acid number.

**Table 2 ijms-22-06973-t002:** De novo missense mutations in the human *EEF1A2* gene and corresponding amino acid replacements in eEF1A2 that have been associated with severe neurological disorders, including epilepsy and intellectual disabilities in children [34,86,88,93] and cardiomyopathies [90,91].

Mutation	Amino Acid Change	Mutation	Amino Acid Change
c.46G>C	p.Val16Leu	c.364G>A	p.Glu122Lys
c.49G>C	p.Asp17His	c.370G>A	p.Glu124Lys
c.208G>A	p.Gly70Ser	c.374C>A	p.Ala125Glu
c.211A>C	p.Ile71Leu	c.754G>C	p.Asp252His
c.271G>A	p.Asp91Asn	c.796C>T	p.Arg266Trp
c.274G>A	p.Ala92Thr	c.998C>T	p.Pro333Leu
c.292T>C	p.Phe98Leu	c.1150G>C	p.Gly384Arg
c.293T>G	p.Phe98Cys	c.1267C>T	p.Arg423Cys
c.294C>A	p.Phe98Leu	c.1295C>T	p.Thr432Met
c.304A>G	p.Met102Val *		

* variant of uncertain significance [88].

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
