# Peer review of "On the Need to Tell Apart Fraternal Twins eEF1A1 and eEF1A2, and Their Respective Outfits"

_ijms, 2021, doi:10.3390/ijms22136973_

Round 1

Reviewer 1 Report

This is a very professional and comprehensive review on eEF1A1 and eEF1A2 proteins, showing a high meritorical value.

There are some fields which should be mentioned in this interesting manuscript to show other current important targets for investigation of these proteins, such as neurodegeneration - Alzheimer's disease, Parkinson's disease, as well as possible COVID treatment with Plitidepsin.

Author Response

>>There are some fields which should be mentioned in this interesting manuscript to show other current important targets for investigation of these proteins, such as neurodegeneration - Alzheimer's disease, Parkinson's disease, as well as possible COVID treatment with Plitidepsin.

A possible COVID treatment with Plitidepsin, which we mentioned in doi 10.1002/cbic.202000516, is not really in line with the subject of this special issue. As regards eEF1A2's involvement in neurons, an extra paragraph has been added at the introduction, but a distinct involvement of eEF1A1 and eEF1A2 proteins in Alzheimer's or Parkinson's disease is not firmly established in the literature, with most reports dealing with model systems in other species.

Reviewer 2 Report

In this review, Mills and Gago pointed out differences between two paralogous proteins: eEF1A1 and eEF1A2 that have long been described as eEF1A. eEF1A2 was actually discovered with specific mutations that leads to genetic disorders. The authors have analyzed the structural differences and a large part of the manuscript is devoted to their post-translational modifications that add a layer of complexity. This is a very interesting review (last update on eEF1A was in 2015) in which the authors showed clearly this potential functional complexity that exists between the two proteins.

This review remains difficult to read especially when the reader is not specialist in protein structure. A lot of data comes from crystallography and mass spectr. analyzes. This should be specified in the abstract. Some reorganization of the manuscript could help make it clearer, especially at the beginning. Sections 1 and 2 can be grouped together in which the authors introduce the two proteins (general characteristics), their function in translation (actually it is divided into two), then expand to other functions and the problem of discrimination between eEF1A1 and eEF1A2? They should introduce all the different parts of the review in the 1st section. In the same order, section 5 could be included in section 6.

I encourage the authors to better illustrate, when possible, each section with even simple figures. Figure 2 could show the GTP conformation to highlight the interaction between Asp110 and Arg240 but not in the dimeric conformation. I would expect to see the structure of both proteins and not only eEF1A2 to see differences… Many amino acids are described in the manuscript and some could be shown into figures.

Would it be possible to list in a table all the similarities and differences in the PTMs between the two eEF1A?

Minor comments

Some abbreviations are only used once or twice and are therefore not really required especially when defined at the beginning of the manuscript and used at the end: e.g. WH2, PBZ, APLF, ARH, IDR…

  1. 29, the translation process is divided into four phases: initiation, elongation, termination and ribosome recycling. Please correct

l.29 to 40 require additional references

l.35 and l.36 require references

l.43 to l.56 are not very related with the article and quite confusing and difficult to understand.

l.107: The authors did not comment that fact that GDP dissociates 7-fold slowly from eEF1A2 than eEF1A1. What is the impact on the translation rate?

l.162, define wst

  1. 422, define 6RA.

l.414 and 525 typing errors

Table 1: The amino acid should be numbered. What are CB5 and HT7 regions? These are not defined.

Fig. 1 boxes are missing, aa at position 197 could be highlighted

Fig.2C is not cited

reference 13 seems to be incomplete

Author Response

>> Some reorganization of the manuscript could help make it clearer, especially at the beginning. Sections 1 and 2 can be grouped together in which the authors introduce the two proteins (general characteristics), their function in translation (actually it is divided into two), then expand to other functions and the problem of discrimination between eEF1A1 and eEF1A2? They should introduce all the different parts of the review in the 1st section. In the same order, section 5 could be included in section 6.

Sections have been regrouped following this valuable recommendation.

>>Figure 2 could show the GTP conformation to highlight the interaction between Asp110 and Arg240 but not in the dimeric conformation. I would expect to see the structure of both proteins and not only eEF1A2 to see differences… Many amino acids are described in the manuscript and some could be shown into figures.

We chose not to complicate the figures in excess and not to reiterate on previously published material, e.g., doi: 10.1002/cbic.202000516. Moreover, additional labels are likely to distract the reader and, from a distance, eEF1A1 and eEF1A2 would look exactly the same. Figure 2, therefore, focuses on the relevant mutations. On the other hand, Figure 3 shows two instances of the monomeric GTP-bound conformation, and a new Figure 4 has been included to account for some very recent findings regarding Ser385 in eEF1A2 (still in press, ref. 107).

>>Some abbreviations are only used once or twice and are therefore not really required especially when defined at the beginning of the manuscript and used at the end: e.g. WH2, PBZ, APLF, ARH, IDR…

We have taken into account this comment, removed the unnecessary ones and changed the corresponding sentences.

l. 29, the translation process is divided into four phases: initiation, elongation, termination and ribosome recycling. Please correct.

The opening sentence has been rewritten as follows: "Protein synthesis is one of the most sophisticated biochemical processes in living cells and comprises the same steps in eukarya, bacteria and archaea: initiation, elongation, and termination─ribosome recycling notwithstanding. "

l.29 to 40 require additional references: the following two references have been added

1. Riis, B.; Rattan, S.I.; Clark, B.F.; Merrick, W.C. Eukaryotic protein elongation factors. Trends Biochem. Sci 1990, 15, 420-424, doi:10.1016/0968-0004(90)90279-k.

2. Andersen, G.R.; Nissen, P.; Nyborg, J. Elongation factors in protein biosynthesis. Trends Biochem. Sci 2003, 28, 434-441, doi:10.1016/S0968-0004(03)00162-2.

l.35 and l.36 require references: the following two references have been added

3. Andersen, G.R.; Pedersen, L.; Valente, L.; Chatterjee, I.; Kinzy, T.G.; Kjeldgaard, M.; Nyborg, J. Structural basis for nucleotide exchange and competition with tRNA in the yeast elongation factor complex eEF1A:eEF1Ba. Mol. Cell 2000, 6, 1261-1266.

4. Andersen, G.R.; Valente, L.; Pedersen, L.; Kinzy, T.G.; Nyborg, J. Crystal structures of nucleotide exchange intermediates in the eEF1A-eEF1Ba complex. Nat. Struct. Biol. 2001, 8, 531-534, doi:10.1038/88598.

l.107: The authors did not comment that fact that GDP dissociates 7-fold slowly from eEF1A2 than eEF1A1. What is the impact on the translation rate?

This issue, which is not related to the subject under discussion, has been dealt with by Crepin et al. and Carriles et al. Nonetheless, we have incorporated a new paragraph(l. 431-440) dealing with a related aspect:

Loss of METTL13 function alters translation dynamics and results in changed translation rates of specific codons [126], but other possible biochemical and functional consequences have not been explored in comparable detail. In fact, the side 433 chain of Lys55 directly interacts with the indole ring of Trp196 in both GTP- [127] and GDP-bound forms [58,59] of eEF1A1/2, but only in the GDP-bound dimeric state does it interact with the phenol of Tyr56. Since we have pointed out that motions of Phe54, Tyr56 436 and Trp196 appear to be involved in nucleotide binding and exit [58], it is therefore conceivable that dimethylation of Lys55 serves the purpose of strengthening cation-pi interactions [128] with the aromatic residues of Trp196, Tyr56, or even the neighboring Tyr29, to stabilize a particular protein conformation.

l. 162: wst (for 'wasted') has been defined

l. 422, define 6RA: PDB entry 6RA9

l. 414 and 525 typing errors: corrected

>>Table 1: The amino acid should be numbered. What are CB5 and HT7 regions? These are not defined.

Those were the original peptide names. They have been removed, and amino acids have been numbered.

>>Fig. 1 boxes are missing, aa at position 197 could be highlighted.

This has been done and the boxes are clearer now (earlier lost in the PDF conversion because of the white color) because we converted the text into an image.

>>Fig. 2C is not cited

The reviewer probably means Fig. 3C, which is now cited below the figure.

>>reference 13 seems to be incomplete

The reference to this electronic book chapter has been completed.